# Proximal Tibiofibular Dislocation in a Closing-Wedge High Tibial Osteotomy Causes Lateral Radiological Gapping of the Knee: A Prospective Randomized Study

**DOI:** 10.3390/jcm9061622

**Published:** 2020-05-27

**Authors:** Raúl Torres-Claramunt, Juan Francisco Sánchez-Soler, Pedro Hinarejos, Aleix Sala-Pujals, Joan Leal-Blanquet, Joan Carles Monllau

**Affiliations:** 1Orthopaedic Department, Hospital del Mar, Universitat Autònoma Barcelona, 08003 Barcelona, Spain; 96705@parcdesalutmar.cat (R.T.-C.); phinarejos@parcdesalutmar.cat (P.H.); 61320@parcdesalutmar.cat (A.S.-P.); 87024@parcdesalutmar.cat (J.C.M.); 2IMIM (Hospital del Mar Medical Research Institute), 08003 Barcelona, Spain; 3Orthopaedic Department, ICATME-Institut Universitari Quirón-Dexeus, Universitat Autònoma Barcelona, Sabino de Arana 5-19, 08017 Barcelona, Spain; 4Orthopaedic Department, Hospital de Igualada, Consorci Sanitari de l’Anoia, 08700 Barcelona, Spain; jlealblanquet@gmail.com

**Keywords:** closing-wedge osteotomy, knee stability, stress radiology, knee, high tibial osteotomy

## Abstract

Background: To determine whether a proximal tibiofibular joint dislocation (TFJD) increases lateral compartment gapping more than a fibular head osteotomy (FHO) during a closing-wedge high tibial osteotomy (CWHTO). The second objective was to determine whether lateral compartment gapping affects clinical outcomes. Methods: A prospective randomized clinical study was carried out that included 18 patients in Group 1 (FHO) and 18 in Group 2 (TFJD). Varus-stress radiographs of all the patients with both knees at full extension and at 30° of flexion were studied pre-operatively and 12 months post-operatively. Lateral compartment gapping was measured in millimeters. The Knee Society Score (KSS) was used to assess clinical stability. Results: The difference between the pre- and post-operative measurements relative to gapping in the lateral knee compartment at 0° of knee flexion was 1.3 mm (SD 1.8) in Group 1 and 4.5 mm (SD 2.4) in Group 2 (*p* = 0.006). At 30° of knee flexion, this difference was 1.9 mm (SD 1.2) in Group 1 and 5.2 mm (SD 3.1) in Group 2 (*p* = 0.01). No differences were observed in the pre- and post-operative period relative to gapping in healthy knees. Pre-operatively, both groups presented similar KSS knee values: Group 1 with 54.7 (SD 11.7), Group 2 with 54.8 (SD 11.1) (n.s.). Post-operatively, these values were also similar: Group 1 with 93.2 (SD 7.4), Group 2 with 93.5 (SD 5.5) (n.s.). Conclusions: In patients who have undergone a CWHTO, TFJ dislocation increases knee lateral compartment gapping when compared to an FHO at 0° and 30° of knee flexion. However, this fact seems to have no repercussion on the functional status of the knees as measured with the KSS at the one-year follow-up.

## 1. Introduction

A high tibial osteotomy (HTO) is considered an effective treatment option for varus knee patients with medial knee osteoarthritis [1,2,3]. The transfer of weightbearing forces to an unaffected lateral knee compartment relieves the pain in the unloaded medial compartment. Classically, a lateral closing-wedge (CW) HTO or the so-called Coventry procedure was considered the gold standard for patients with successful results [1,2,3,4,5]. However, in the last decade, the medial open-wedge HTO seems to have overcome CWHTO in terms of the number of cases in which it was used to treat symptomatic varus knees. However, a meta-analysis published in the Cochrane database showed no differences between the two procedures [1]. Moreover, a recent paper reported even better results with the CWHTO [4].

Tibiofibular joint (TFJ) manipulation is considered one of the disadvantages of the CWHTO in comparison to the open-wedge HTO. Basically, there are two surgical options described to manage the TFJ upon closing the osteotomy: TFJ dislocation and fibular osteotomy. The choice between these two options is mostly based on the surgeon’s preference. Although some authors prefer to do the fibular osteotomy in the mid-third of the fibula, the original technique describes this surgical gesture being done to the fibular head [6]. Palsy of the peroneal nerve is a potential complication after a proximal fibular head osteotomy (FHO) [7]. However, the fact that the rate of non-union is greater when the osteotomy is done in the mid-third of the fibula must be considered [8].

In recent years, the anatomy and the biomechanics of the posterolateral part of the knee was studied in detail [9,10,11,12,13,14,15,16,17]. The lateral collateral ligament (LCL) and the popliteus-fibular ligament (PFL) are two structures that have their origin in the fibular head and act on lateral knee stability. Some of these studies highlight the importance of an anatomical reconstruction of these ligaments when lateral knee instability is present. They also cite the importance of correct graft tensioning in achieving good stability [13].

Therefore, manipulation of the TFJ during a CWHTO may have repercussions on the lateral stability of the knee due the LCL and PFL having their origin in this area. TFJ dislocation along with ascending fibular head displacement and the consequence of ligament tensioning may alter the lateral stability of the knee more than the fibular head osteotomy. However, as far as we know, this has not been studied until now.

The aim of this study is to determine whether two different ways of surgically managing the proximal TFJ have any radiological and/or clinical repercussions on lateral knee stability in those patients who had undergone a CWHTO. The main hypothesis of this study is that a TFJ dislocation increases lateral compartment gapping more than an FHO. A second hypothesis is that this fact has limited clinical repercussions.

## 2. Materials and Methods

A prospective randomized study that initially included 40 consecutive patients with medial knee pain was proposed using CWHTO to correct a varus knee. The diagnoses included isolated medial compartment degenerative arthritis and medial post-meniscectomy syndrome. Patients with a post-traumatic varus knee, or those patients that had previous anterior-posterior or medial-lateral knee instability, were excluded from the study. Patients who had previously undergone a surgery on the knee under study, except for those that had had a simple arthroscopy, were also excluded. All of these patients were previously unsuccessfully managed with conservative treatment. Four patients were excluded from the study: one for previous medio-lateral instability and 3 had had a previous surgery on the involved knee. Finally, 36 patients were included. All the surgeries were performed in a single center by 4 expert knee surgeons. The study was approved by the ethics committee of our institution (2015/6527/l). Patients signed informed consent to participate in the study.

### 2.1. Surgical Procedure and Randomization

An oblique incision was performed in the anterolateral part of the tibial plateau. Before the tibial osteotomy, the surgical procedure performed on the TFJ was randomized. Group 1 was assigned to undergo an FHO and Group 2 was assigned to undergo a TFJ dislocation. The randomization list was computer-generated. The CWHTO procedure was performed with the Natural-Knee^®^ High Tibial Osteotomy (HTO) System (Zimmer^®^, Warsaw, IN, USA). In Group 1, a resection of the medial third of the fibular head was performed using a saw and an osteotome. In Group 2, a transection of the proximal tibiofibular ligaments was performed using a blunt osteotome (Figure 1). The osteotomy was fixed with two 6.5 mm cancellous screws in the tibial epiphysis and three 4.5 mm cortical screws in the metaphysis in all the cases. Both groups were comparable in terms of age, sex, body mass index, previous femorotibial angle and number of corrected degrees (Table 1). The mean femorotibial angle in healthy knees was 176.9 degrees (SD 1.7).

All the patients followed the same rehabilitation protocol. Continuous passive movement and partial weightbearing were allowed the first post-operative day. Full weightbearing was allowed from the 3rd post-operative week.

### 2.2. Radiological Assessment

A pre-operative radiological study was performed on all the patients. It included an antero-posterior and a lateral view, a Rosenberg view and a weightbearing full-leg-length X-ray. The mechanical axis of the limb was drawn from the hip to the ankle in a radiograph. The new mechanical axis was set 62.5% laterally from the medial edge of the tibial plateau [18]. It corresponded to a valgus angulation of 2° to 4°. All the radiological studies were analyzed by using the PACS computer system (Picture Archiving and Communication System).

The lateral stress radiograph was performed at full extension and 30° of flexion on both knees while applying a 15 KPa load with a TELOS-stress device. The gapping distance in the lateral compartment was defined as the closest perpendicular distance (in millimeters) between the central aspect of the lateral femoral condyle and the corresponding lateral tibial plateau (Figure 2). For this purpose, the thickness of the articular cartilage surface was not considered [12]. An expert orthopedic radiologist who was not involved in the study did these measurements pre-operatively and 1 year post-operatively.

### 2.3. Clinical Assessment

Functional assessment was carried out with Knee Society Score (KSS) [19] in its Spanish version [20]. This score was split into 2 subscales: knee and function. The knee section was used to assess knee stability. This score was administered pre-operatively and 12 months post-operatively.

### 2.4. Statistical Analysis

An a priori sample size was calculated. To detect a difference of 2 mm in both groups with α = 0.05 and a power of 80%, 18 patients were needed in each group. A maximum patient loss of 10% was considered.

The mean and standard deviations were calculated for each continuous variable. The results were statistically analyzed and compared using the Student’s t-test for parametric data with a normal distribution. The SPSS program was used for the statistical analysis and a *p*-value of 0.05 was considered statistically significant.

## 3. Results

All the patients were assessed pre-operatively and one year after the surgery. There was no deep infection or deep venous thrombosis and there were no neurological complications. No further surgeries were done on the knee in any of the patients during the follow-up year.

Table 2 shows the lateral knee compartment gapping (in mm). No differences were found between the two groups with both knees (healthy and operated) at 0° and 30° of flexion in either period (pre- or post-operative). Table 3 shows the gapping increment after the CWHTO for both groups as well as the values for the healthy knee. While the gapping increment was minimal in Group 1, Group 2 saw a 4.5 mm (SD 2.4) increment at full extension and a 5.2 mm (SD 3.1) increment at 30° of knee flexion.

Table 4 shows the clinical and functional results for both groups. With regard to the KSS, no differences were observed in the groups studied in either the pre-operative or post-operative period.

## 4. Discussion

The main finding of this study is that in a varus-stress radiograph, lateral knee compartment gapping is greater when a TFJ dislocation is done instead of an FHO to close the CWHTO. This finding confirms the main hypothesis of the study. However, a second finding of the study is that this fact does not translate into worse clinical or functional outcomes as measured with the KSS at the one-year follow-up.

Jacobsen et al. [21] were the first to study varus-stress radiographs in healthy patients. They observed that lateral compartment gapping was 9.2 mm in normal knees, with a side-to-side difference less than 2 mm. They concluded that a gapping increment of 2 mm implied an injury of the lateral ligaments of the knee. More recently, Yoo et al. [22] studied the clinical opening of the knee compartment in a healthy population with a custom-made device. They determined that this gapping was 9.3 mm (range 5.1–13.6) in males and 9.1 mm (range 5.1–11.9) in females. These results were similar to the ones found in healthy knees in this study. Laprade et al. [12], in a cadaveric study, observed that normal gapping in an intact knee with a 12-Nm varus-stress load was 8.8 mm. They concluded that clinicians should suspect an isolated tear of the lateral collateral ligament when there is an increase of 2.7 mm in lateral gapping in a clinician-applied varus-stress radiograph. A grade-III posterolateral corner injury should be considered if the values increase to 4 mm. The results obtained in the TFJ dislocation group are greater than 4 mm at full extension and 30° of knee flexion in this study. Therefore, according to Laprade et al. [12], these gapping values would be equivalent to a grade-III posterolateral corner injury.

Based on the International Knee Documentation Committee’s [23,24] suggestions, lateral knee compartment stability was assessed with a varus-stress radiograph with the knee at 20° to 30° of flexion [11,12,13,25]. However, it would have been better to perform the study at both 0° and 30° of knee flexion to fully assess varus stability. It seems that increased gapping at full extension translates into an injury of the lateral collateral ligament and a possible injury of the posterior cruciate ligament [26]. However, increased gapping at 30° of knee flexion suggests an injury of the lateral collateral ligament with the possible association of the posterolateral structures of the knee. Although the opening at 30° of knee flexion is greater than at 0° of knee flexion in this series, this difference is not statistically relevant. It could be hypothesized that this minimal difference is due to the fibular head rise after dislocation, while the lateral capsular, as well as the cruciate ligaments, remain intact.

This is the first series that assesses lateral instability in patients who have undergone an HTO. In this series, the increased lateral instability, objectified in a stress radiograph, has no clinical repercussions when measured with the KSS. We should keep in mind that the KSS is not a sensitive enough tool to detect a clinical difference due to varus instability. In fact, only 15 out of 100 points in the knee section of the KSS are dependent on varus-valgus stability. However, no score analyzes varus-valgus stability with precision. The mean age of the series presented here is 53 years-old and all the patients who underwent this procedure presented degenerative changes in the medial compartment of the knee. For this reason, the KSS was chosen to assess the functional outcomes after the CWHTO instead of another questionnaire.

Other reasoning that might explain why objective lateral instability does not translate into a worse clinical outcome is that the objective Fujisawa point is lateral enough for the axial load in daily activities to translate into clinical differences. The LCL is insufficient but instability is not recognized in valgus loads.

A limitation of this study may be the high mean age of patients with low functional demand and with degenerative changes in the medial compartment. This fact can explain the radiological instability without finding clinical repercussions. Another limitation is the short follow-up time for clinical findings. The suggestion is that these radiological findings could translate into symptomatic clinical instability in a younger population or if longer-term functional outcomes were analyzed, especially in patients with hypo-correction who continue loading in varus.

One conclusion is that at full extension and 30° of knee flexion, the TFD during a CWHTO increases lateral knee compartment gapping when compared to an FHO. The second conclusion of the current study is that this radiological instability is not translated into clinical lateral instability when assessed with the KSS at the one-year follow-up.

## Figures and Tables

**Figure 1 jcm-09-01622-f001:**
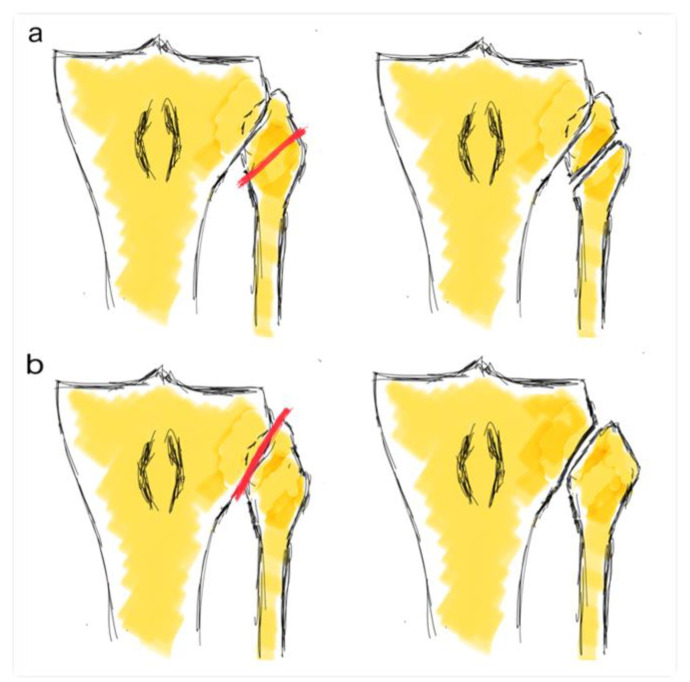
Fibula head osteotomy (**a**) and tibiofibular joint dislocation (**b**).

**Figure 2 jcm-09-01622-f002:**
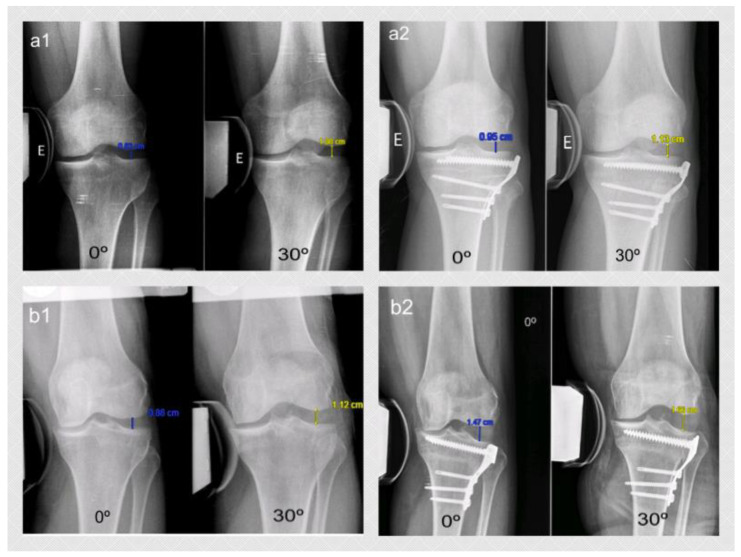
Varus-stress radiological image at full extension and 30° of knee flexion in a patient randomized to Group 1 (**a1** pre-op; **a2** post-op) and in Group 2 (**b1** pre-op; **b2** post-op). The measurement was done between the central aspect of the lateral femoral condyle and the corresponding lateral tibial plateau.

**Table 1 jcm-09-01622-t001:** Demographic data, preoperative femorotibial (FT) angle and number of corrected degrees in both groups. Not significant (n.s.)

	Group 1	Group 2	*p*-Value
Age (years)	52.6 (SD 10.1)	53.3 (SD 7.7)	n.s.
Gender (male/female)	13/5	14/4	n.s.
Body mass index (kg/m^2^)	27.6 (SD 3.9)	27.2 (SD 4.2)	n.s.
Side (right/left)	11/7	8/10	n.s.
Previous FTA	173.57 (SD 1.9)	173.2 (SD 2.5)	n.s.
Number of corrected degrees	10° (SD 2.5)	8.7° (SD 2.3)	n.s.

**Table 2 jcm-09-01622-t002:** Lateral knee compartment gapping (in millimeters) in a varus-stress radiograph in the operated and the healthy knees for both measurements: 0° and 30° of knee flexion. Not significant (n.s.)

	Pre-operatively			Post-operatively		
**Operated Knee**						
	**Group 1**	**Group 2**	***p*-Value**	**Group 1**	**Group 2**	***p*-Value**
0°	10.8 (SD 2.1)	8.9 (SD 2.3)	n.s.	12.1 (SD 2.3)	13.4 (SD 3.2)	n.s.
30°	10.6 (SD 2.6)	9.6 (SD 2.4)	n.s.	12.5 (SD 1.6)	14.8 (SD 3.4)	n.s.
**Healthy Knee**						
	**Group 1**	**Group 2**	***p*-Value**	**Group 1**	**Group 2**	***p*-Value**
0°	9.9 (SD 3)	9.5 (SD 2.8)	n.s.	10.3 (SD 1.9)	9.6 (SD 2.3)	n.s.
30°	10.1 (SD 2.1)	11.1 (SD 3.5)	n.s.	9.9 (SD 2.2)	10.9 (SD 3.1)	n.s.

**Table 3 jcm-09-01622-t003:** Difference (in mm) between the pre- and post-operative measurements with regard to the lateral knee compartment gapping in varus-stress radiographs with both knees (operated and healthy) at full extension and 30° of flexion. Not significant (n.s.)

Operated Knee				Healthy Knee			
	Group 1	Group 2	*p*-Value		Group 1	Group 2	*p*-Value
0°	1.3 (SD 1.8)	4.5 (SD 2.4)	0.006	0°	0.4 (SD 1.4)	0.1 (SD 1.5)	n.s.
30°	1.9 (SD 1.2)	5.2 (SD 3.1)	0.01	30°	0.2 (SD 1.2)	0.2 (SD 1.7)	n.s.

**Table 4 jcm-09-01622-t004:** Knee Society Score (KSS) values in the pre-operative and post-operative period. Not significant (n.s.)

	Pre-operative			Post-operative		
	Group 1	Group 2	*p*-Value	Group 1	Group 2	*p*-Value
KSS knee	54.8 (SD 11.1)	54.7 (SD 11.7)	n.s.	93.5 (SD 5.5)	93.2 (SD 7.4)	n.s.
KSS function	71.7 (SD 5.2)	68.4 (SD 8.1)	n.s.	92.8 (SD 7.5)	90.3 (SD 11.4)	n.s.

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
