# Peer review of "Proximal Tibiofibular Dislocation in a Closing-Wedge High Tibial Osteotomy Causes Lateral Radiological Gapping of the Knee: A Prospective Randomized Study"

_jcm, 2020, doi:10.3390/jcm9061622_

Round 1

Reviewer 1 Report

Dear Authors, 

this paper is a perspective study about an interesting issue, namely the role of Tibiofibular joint (TFJ) manipulation in the close-wedge high tibial osteotomy (HTO) in comparison to the open-wedge HTO.

This paper is well written, clearly expose and very well structured. The issue covered is fascinating and very common in the daily-practice activity.

This paper could help orthopaedics to manage more adequately the TFJ during a close-wedge HTO.

Author Response

English checked. Thanks for your review

Reviewer 2 Report

Based on varus-stress radiographs of 36 patients at full extension and at 30 degrees of flexion, the authors show that CWHTO followed by TFJ dislocation leads to a larger "lateral knee compartment gapping" than a CWHTO followed by an FHO.

The authors also show that increased "lateral knee compartment gapping" - although correlated with an increased "radiological lateral instability" of the knee when assessed with stress radiographs - cannot be confirmed by clinical lateral instability when assessed by the KSS.

With their work the authors aim to confirm a hypothesis that "TFJ dislocation along with ascending fibular head displacement and the consequence of ligament tensioning may alter the lateral stability of the knee more than the fibular head osteotomy."

Since hypothesis and conclusion do contain an implicit causal chain, namely that an enlarged lateral joint space when assessed with stress radiographs is equated with lateral (radiological) instability, it seems to me that the title of the article may be misleading. In fact, you compare results of CWHTO+TFJD with results of CWHTO+FHO in view of the lateral joint space as measured in stress radiographs. A conclusion regarding stability cannot be confirmed. Hence, I would advocate a somewhat more objective title instead of the chosen one, which, for example, could read as follows:

"A closing-wedge high tibial osteotomy (in combination with / followed by) a proximal tibiofibular joint dislocation causes larger lateral compartment gapping in stress radiographs than (with / by) a fibula head osteotomy: Evaluation of radiological and clinical instability."

Some additional comments:

Lines 51 and 52: not really "recent" (2014, 2016)

Line 71: consequence

Everywhere: Varus is always written with upper case "V" (why? valgus is not)

Line 96: "to undergo a TFJ dislocation"

Table 1: why is p-value listed here?

Line 117: 62.5% of what?

Line 119: "calculated by using the PACS" does not make any sense. The PACS is an image database system - any calculations are performed using some kind of dedicated software.

Line 141: What of your data had a normal distribution and what a non-parametric distribution?

Author Response

Thanks for your review.

Title. has been modified. We changed the word instability by gapping. It could be more accurate. 

Lines 51 and 52. Delete recent (line 2)

Line 71. Change for consequence

Everywhere. It is true. We changed all the V for v in the word varus. 

Line 96. changed following the reviewer suggestion 

Table 1. This table compared the data obtained in both groups. There were no differences between them. So, both groups were comparable. 

Line 117. 62.5% of what? It is the exact point were the mechanical axis should be stablished after a HTO.  Known as Fujisawa´s point. We think it is not necessary to change the sentence. 

Line 119. The sentence about the PACS has been modified following the reviewer suggestion.

Line 141. After reviewing the analysis with our statistical assistant, only t-student had been finally used, all the data were considered parametric data with normal distribution. This part of methods was written before the analysis and it is already modified
